# Innovative Therapies Targeting the Virus and the Host for Treating Chronic Hepatitis B Virus Infection: From Bench to Bedside

**DOI:** 10.3390/vaccines10050746

**Published:** 2022-05-10

**Authors:** Sheikh Mohammad Fazle Akbar, Mamun Al Mahtab, Sakirul Khan, Osamu Yoshida, Julio Cesar Aguilar, Guillen Nieto Gerardo, Yoichi Hiasa

**Affiliations:** 1Department of Gastroenterology and Metabology, Graduate School of Medicine, Ehime University, Ehime 791-0295, Japan; yoshida.m.ehime@gmail.com (O.Y.); hiasa@m.ehime-u.ac.jp (Y.H.); 2Department of Hepatology, Bangabandhu Sheikh Mujib Medical University, BSMMU, Dhaka 1000, Bangladesh; shwapnil@agni.com; 3Department of Microbiology, Oita University, Oita 879-5593, Japan; sakirul.khan@gmail.com; 4Center for Genetic Engineering and Biotechnology, Havana 10600, Cuba; julio.aguilar@cigb.edu.cu (J.C.A.); gerardo.guillen@cigb.edu.cu (G.N.G.)

**Keywords:** chronic hepatitis B, antiviral therapy, immune modulators, innovative treatment

## Abstract

Chronic hepatitis B (CHB) is a highly complicated pathological process in which the disease is initiated by the hepatitis B virus (HBV); however, host immune responses are primarily responsible for variable extents of liver damage. If the patients with CHB remain untreated, many CHB patients will eventually develop complications like cirrhosis of the liver (LC) and hepatocellular carcinoma (HCC). In 2019, an estimated 882,000 patients died due to HBV-related complications worldwide. Accordingly, several drugs with antiviral properties have been used to treat CHB patients during the last four decades. However, the treatment outcome is not satisfactory because viral suppression is not usually related to the containment of progressive liver damage. Although proper reconstruction of host immunity is essential in CHB patients, as of today, there is no acceptable immune therapeutic protocol for them. These realities have exposed new, novel, and innovative therapeutic regimens for the management of CHB patients. This review will update the scope and limitation of the different innovative antiviral and immune therapeutic approaches for restoring effective host immunity and containing the virus in CHB patients to block progression to LC and HCC.

## 1. Introduction

Hepatitis B virus (HBV) is a DNA virus and can infect humans of all ages. The pathogenesis of HBV is highly complex and versatile. HBV has infected about 2 billion people in the world at some point in their life, but about 85% of the HBV-infected persons effectively control the infection. Only about 15% of the total HBV-infected subjects (about 296 million) develop chronic infection. In the scenario of chronic HBV infection, the virus persists in the infected host for more than six months after the initial infection, usually expresses hepatitis B surface antigen (HBsAg), and exhibits HBV DNA in their blood. Accordingly, these chronic HBV-infected subjects are permanent and living reservoirs of HBV infection and transmit the virus to healthy HBV-uninfected subjects [1,2,3]. A certain percentage of these chronic HBV-infected subjects develop features of hepatic inflammation and complications such as cirrhosis of the liver (LC), hepatic failure, and hepatocellular carcinoma (HCC). According to the World Health Organization (WHO), about 882,000 patients died due to HBV-related liver diseases in 2019 [1].

Almost all chronic HBV-infected patients usually develop exacerbation and remission of hepatitis for decades before progressing to LC and HCC. Once an HBV-infected person develops LC and HCC, there is no curative therapy for these pathological processes, and the patients with LC and HCC usually follow downhill courses. These impose considerable burdens on the quality of life of the patients and the health care delivery systems of the country. However, HBV-related complications and deaths, mainly caused by LC and HCC, may be contained if chronic HBV-infected subjects can be treated appropriately when they pass through the stage of chronic hepatitis B (CHB) that is characterized by the presence of HBsAg and HBV DNA in the blood with variable levels of liver damage, shown by the elevation of alanine aminotransferase (ALT) in the blood [4,5]. The CHB stage of the pathological process of chronic HBV infection also usually continues for decades, allowing sufficient time for diagnosis and treatment.

To treat CHB patients, various antiviral drugs have been developed during the last four decades. Initially, CHB patients have been treated by interferon (IFN) and later by their pegylated forms [6]. With the advent of science, a group of oral antiviral drugs, nucleos(t)ide analogs (NUCs, lamivudine, adefovir, entecavir, telbivudine, and tenofovir) have been recommended for CHB patients from 1990s [7]. As of now, NUCs are mainly used for the treatment of CHB patients as there are several positive aspects of using NUCs. The use of NUCs usually drastically reduces HBV DNA in the blood, and several CHB patients also become HBV DNA negative due to the usage of NUCs. NUCs have been reported to contain liver damage to certain levels in some CHB patients [8]. Clinically, the hepatotoxicity of NUCs is negligible. However, the therapeutic efficacy of NUCs is not completely satisfactory. Also, NUCs should be used for a prolonged period, and cessation of NUCs administration is related to the reappearance of HBV DNA in CHB patients [9,10,11]. This induces severe complications like hepatic failure, which may be life-threatening in many patients [10]. Taken together, chronic HBV infection is responsible for about one million deaths due to the lack of proper drugs to tackle this pathological situation, and, they are permanent reservoirs of HBV. These limitations indicate a pressing and emergent need to develop innovative, evidence-based, patient-friendly, safe, and efficacious medicines for CHB patients. Several new drugs targeting the virus (HBV) and host immunity have been developed in this line. The present review will provide a bird’s eye view of the scope of limitations of these innovative therapeutic approaches. Finally, a sketch regarding the possible mode of an effective therapeutic regimen for CHB will be provided.

## 2. The Fundamental Design of Novel Therapeutics for CHB: Targeting Both the Virus and the Host Immunity

HBV is a hepatotropic virus and usually causes either a self-limiting infection with almost complete recovery or an asymptomatic infection or develops a chronic infection that may progress to complications like LC or HCC. The virus is mainly non-cytopathic and cannot induce inflammation (hepatitis). The virus is not also an inducer of hepatic fibrosis. Various investigations have shown that the inflammation, fibrosis, and hepatocellular carcinoma are not due to the direct effect of HBV, a non-cytopathic virus. Instead, the host immunity is dysfunctional and dysregulated in CHB patients, which ultimately induces inflammation of hepatitis, progression of hepatic fibrosis, and genesis of hepatocarcinogenesis [12,13,14].

Thus, one of the main targets of treatment of CHB patients is to completely eradicate all forms of HBV from the infected hepatocytes so that virus-induced immune dysfunction can be omitted. However, complete eradication of HBV from CHB patients seems to be an unachievable goal due to the highly complex nature of HBV infection in the human hepatocytes. Infection with HBV leads to the production of covalently closed circular DNA (cccDNA) in the nucleus of the hepatocytes as episomal macrochromosomes. Most of the commercially available antiviral drugs are unable to eradicate or contain cccDNA satisfactorily. Thus, innovative drugs should be developed to control cccDNA to a reasonable level [15,16].

At the same time, the dysregulated and dysfunctional immunity of the hosts in CHB patients should be optimized for favoring the containment of liver damage. The notable role of host immunity became evident from the fact that although HBV infects about 2 billion people in the world, only an estimated 296 million become chronic HBV-infected. In addition, only a certain percentage (estimated 15–25%) of the total 296 million chronically HBV-infected patients develop CHB (a condition that allows replication of HBV with damage to the liver). Thus, in most HBV-infected personnel, HBV is efficiently controlled by host immunity, whereas host immunity fails to contain HBV replication and liver damage in some patients with CHB [1].

Thus, a critical analysis of the pathogenesis of CHB versus drug development indicates that efficient drug development for CHB is dependent on (1) innovative therapy capable of controlling different forms of the HBV and their replication cycle, and (2) innovative approaches for shaping the host immunity in CHB patients.

## 3. Innovative Therapy Targeting the Virus to Control HBV Replication

When the life cycle of HBV in the hepatocytes is analyzed with importance on different stages of HBV replication, the pathways shown in Table 1 are of immense significance for replication of HBV and formation of cccDNA. As shown in Table 1, out of these five variables, commercially available antiviral drugs have only provided attention to variable number 4 (Role of HBV polymerase) (shown in blue). NUCs, which are the central pillar of treating CHB at present, are dependent on blocking DNA polymerase activity.

### 3.1. Entry Inhibitor

HBV enters the hepatocytes in multiple steps. This starts with low-affinity viral attachment to hepatocytes. This is followed by subsequent high-affinity interaction with a specific receptor. These processes lead to the internalization of the virus into the hepatocytes [17]. The primary receptor for these processes is sodium taurocholate transporting polypeptide (NTCP). Several innovative agents that target NTCP have been documented. These include myrcludex-B, bile acids, cyclosporins, ezetimibe, ritonavir, irbesartan, and vanitaracin. Among these agents, myrcludex has been used in a phase IIa study to treat hepatitis delta virus infection where HBV also infected the patients. At a higher dose (10 mg), HBV DNA reduction of more than 1 log was recorded in 6 of 10 HBeAg-negative CHB patients receiving myrcludex [18]. Also, this drug was used with pegylated interferon or NUCs in CHB patients. Myrcludex-B has been approved with the name of Bulevertide for medical use in the European Union in July 2020 for patients with chronic hepatitis B/D coinfection [19]. The overall outcome state that it has potent action in inhibiting the entry of the virus, but the capacity of the drug to induce HBsAg negativity or HBsAg reduction was not encouraging [20]. Also, it remains to be shown if the effects of entry inhibitors are sustained or not. Studies are going to on assess these effects by entry inhibitors.

### 3.2. Inhibitors of cccDNA

Although almost all currently available antiviral drugs can reduce or completely block the production of replicative HBV DNA, they are endowed with minimal capacity to contain cccDNA. cccDNA represents a template for viral transcription. Thus, if cccDNA can be eradicated, a functional cure and possibly eradication of HBV may be possible. As of today, several potential inhibitors of cccDNA, such as direct gene editors, epigenetic modifiers, and DNA destabilizers, are in development stages. These drugs are yet to be checked in clinical trials to assess safety and efficacy in CHB patients.

### 3.3. Core Protein Allosteric Modulators

The core protein is essential for HBV pgRNA packaging and reveres transcription, a pathway that must be followed for HBV replication. Core protein allostimulator (CpAM) molecules attack viruses and induce incorrect assembly [21]. Thus, the life cycle of HBV is interrupted by causing an interruption in assembly. This interrupts the life cycle of the virus. Recently, an experimental drug with CpAM activity, named NVR 3-778, has been developed. The drug molecule effectively blocks the formation of HBV DNA; however, the viral rebound was observed after the cessation of drug treatment [22]. JNJ-6379 is another CpAM, which is a phase I study that revealed the therapeutic efficacy to reduce HBV DNA [23]. However, the effect of JNJ-6379 on HBsAg levels is yet to be shown. Another drug of the same feature, ABI-H0731 (vebicovir) has been used as part of combination therapy in patients with CHB with a commercially available antiviral drug. The antiviral effect of this drug is evident when used with other known antiviral drugs, but the impact of HBsAg and HBeAg seroconversion needs to be assessed in a long-term follow-up study.

### 3.4. RNA Interference

RNA interference (RNAi) is able to target HBV transcripts directly and may induce their degradation. Si RNA is able to reduce HBsAg and thus may allow for better restoration of immunity [24]. The role of ARC-520, a drug with RNA interference property, has been checked in CHB patients. It was able to reduce HBV DNA and HBsAg in CHB patients [25]. A modified RNAi, JNJ 3989, revealed antiviral properties and HBsAg reduction in CHB patients [26]. However, these initial studies should be confirmed by long-term follow-up.

### 3.5. Summation of Innovative Therapy for CHB Targeting the Virus

The experience of about four decades with IFNs and NUCs provides the impression that other innovative drugs for containment of HBV replication are yet to be discovered, and their utility should be checked in CHB patients. In line with this, entry inhibitors, core protein allosteric modulators, drugs causing RNA interference, and inhibitors of cccDNA have been developed and tested in CHB patients as innovative drugs targeting the virus HBV. However, more works, especially phase I/II/ and III clinical trials with these innovative drugs, should be accomplished. Also, it seems that combination therapy of two or more antivirals may be an option. Some innovative drugs have been used with commercially available antiviral drugs in small pilot studies with encouraging outcomes. Finally, there is a need to assess the immune-modulatory capacities of these innovative drugs as the dysfunctional and dysregulated immunity of CHB patients’ needs to be handled. These studies would provide an impression that these innovative drugs targeting HBV may be used as an independent therapy, or they should be used as a part of other combination therapy.

## 4. Innovative Immune Therapy for Treating CHB

Although NUCs induced complete negativity of HBV DNA and seroconversion to anti-HBe with an optimum improvement of liver damage in CHB patients, the interim analysis indicated that the role of the antiviral drug is limited for treating CHB. Prevailing scientific evidence and cellular mechanisms underlying the pathogenesis of CHB indicated a vital role of host immunity in CHB patients [27,28,29]. Based on these observations, the treatment of CHB patients using immune modulators started in the early 1990s. However, there has been no generally accepted recommended immune therapy for treating CHB patients till now. Thus, as of today, all sorts of immune treatments are innovative in nature. With the advent of new developments regarding cellular and molecular mechanisms underlying host immunity and the mode of their modulation, some drastic alterations have also been made in immune therapeutic approaches for CHB patients.

### 4.1. Immune Therapeutic Treatment of CHB Patients by Polyclonal Immune Modulators

During the 1980s, it was assumed that the HBV and the immune response of the HBV to the host are essential determinants of CHB pathogenesis. Experimental evidence and clinical data in the early 1990s provided more concrete and acceptable understandings of the cellular and molecular mechanisms underlying this. At that time, it was considered that the host’s immunity was decreased due to chronic HBV infection. With this postulation, the innovative therapy of initial states in the 1980s started with the use of polyclonal immune modulators that activate host immunity. Most of these agents were available at good manufacturing practice (GMP) levels and could be used easily in patients with CHB without reasonable experimentations in animal models of HBV. Other factors that allowed the usage of polyclonal immune modulators are related to the fact that some of these agents have been used in other pathological conditions, and thus there was not any severe safety concern. As shown in Table 2, several immune modulators were used in CHB patients to accentuate their immune status and retrieve a treatment goal.

Among various polyclonal immune modulators, the notable agents are interleukin (IL)-2, IL-12, granulocyte-macrophage colony-stimulating factor, levamisole, alpha galactosylceramide, thymosin alpha, and several other immune modulators. Most of the studies with polyclonal immune modulators in CHB patients were accomplished as a pilot study or clinical trial of phase I type [30,31,32,33,34,35,36,37,38]. Investigators found that these polyclonal immune modulators induce HBV DNA reduction with seronegative hepatitis B antigen (HBeAg) in some patients. Reduction of ALT was also recorded in a certain percentage of CHB patients. However, sustained therapeutic effects of polyclonal immune modulators have not been recorded by any study. All these studies failed to provide data on long-term follow-up in CHB patients. Also, the mechanisms underlying the role of polyclonal immune modulators have not been dissected. Thus, the real implication of polyclonal immune modulators for the treatment of CHB patients was never ascertained and remained to be explored. Neither a phase III study nor dose-dependent investigation with polyclonal immune modulators has been planned so far, although more than three decades have passed after the first clinical trial with these agents for treating CHB patients. As of today, monotherapy with polyclonal immune modulators is not accomplished anymore in CHB patients. However, there remains an opportunity to use these drugs as part of combination therapy with other agents after validation of the preclinical data in animal models of chronic HBV.

### 4.2. Preclinical Studies with HBV Antigen-Based Immune Therapy in Animal Models of Chronic HBV Infection in the Benches and Subsequent Clinical Trials in the Bedsides of CHB Patients

Due to the unsatisfactory outcomes of polyclonal immune modulators for the treatment of CHB patients, the attention of investigators was diverted towards the use of HBV antigen-specific immune therapy in CHB patients. HBV infects humans, and experimental HBV infection can be induced in chimpanzees that mimic the human condition. However, due to the non-availability of chimpanzees and notable scientific and technical limitations with studies using chimpanzees, requisite amounts of preclinical studies with various innovative approaches could not be accomplished in chimpanzees. Due to the advancement of molecular biological techniques, a new and unique animal model of HBV was realized by the mid’1980s. HBV transgenic mice (HBV TM) were produced by microinjecting the HBV genome into the fertilized eggs of mice. Several lines of HBV TM were produced and used for preclinical studies that included insights into HBV pathogenesis and the development of innovative therapies for chronic HBV infection [39,40,41,42]. Some of these HBV TM expressed Dane particles, HBV DNA, HBsAg, and HBeAg. Also, the impaired function of T and B lymphocytes and those of antigen-presenting dendritic cells were evident in HBV TM. This information allowed the usage of HBV TM as an animal model for chronic HBV infection.

At the same time, preclinical studies were done to assess if HBV TM may be used to develop new and novel therapies for chronic HBV infection. Special attention was given if adaptive immunity to HBV-related antigens can be produced in vivo by immunizing HBV TM with HBV-related antigens. Data were accumulated showing that administration of commercially available HBsAg-based vaccine-induced clearance of HBsAg and production of anti-HBs in HBV TM, although all HBV TM harbor large amounts of HBsAg [43]. Thus, the diverse role of endogenous HBsAg versus exogenous HBsAg became evident. Also, some investigators used HBsAg-pulsed dendritic cells to stimulate immunocytes of HBV TM, and these reflected in the clearance of HBsAg from sera of HBV TM [44,45,46]. These data retrieved from studies in HBV TM inspired optimism to initiate clinical trials on patients’ bedsides. Various mods of therapies with HBV antigen-specific immune modulators were accomplished, as shown in Table 3.

### 4.3. HBsAg-Based Vaccine Therapy in Patients with CHB

HBsAg-based immune therapy was initiated in patients with CHB in the mid’1990s. In almost all cases, HBsAg-based preventive, commercially available vaccines were used as innovative immune therapy in CHB patients. The usage of commercially available HBsAg-based vaccines was logical as these have been widely used in humans with considerable safety and preventive efficacy. Several clinical trials were accomplished with the HBsAg-based vaccine in CHB patients. Some studies revealed the reduction of HBV DNA after the end of treatment (EOT) or for 3–6 months with a reduction of serum ALT [47,48,49]. However, others could not show any visible benefit of HBsAg-based therapy as an independent therapy [50,51]. In the final evaluation, it became evident that the sustained therapeutic efficacy of HBsAg-based immune therapy could not be documented by using an HBsAg-based preventive vaccine in CHB. These studies were designed as a pilot study or clinical trial of phase I level. The study design was not assigned to provide long-term follow-up. In addition, most of the studies did not check for the mechanism of action of HBsAg-based immune therapy in CHB patients.

### 4.4. HBsAg plus Anti-HBs Complex Vaccine for CHB

A Chinese group has systematically studied a therapeutic vaccine containing both HBsAg and anti-HBs from mid’ the 1990s [52]. This antigen-antibody complex vaccine was used first in 1995 in CHB patients with some favorable outcomes [53]. Subsequently, a phase II and finally a phase III clinical trial revealed confusing results about this antigen/antibody complex vaccine in CHB patients [54].

### 4.5. Combination Therapy of HBsAg-Based Vaccine, Multi HBV Antigen-Based Vaccine with Antiviral Drugs for Treating CHB

Some investigators postulated that immune therapy with HBV-antigen-based vaccines was not so effective due to the prevalence of high levels of HBsAg and HBV DNA in the sera. High levels of these viral products have long been assumed to have an adverse effect on immune restoration. This initiated another new approach of immune therapy for CHB patients in which antiviral drugs suppressed the virus, and immune therapy was initiated in the microenvironment of low levels of HBV DNA and HBsAg. Different regimens were used to accomplish this trial with immense importance. Some investigators used only HBsAg, whereas others used other surface proteins with small surface antigens. Diverse protocols were also employed. However, the outcome of these trials was not inspiring, and combination therapy of antiviral and HBsAg-based vaccines did not yield a better outcome compared to monotherapy with antiviral drugs [55,56,57,58].

### 4.6. HBsAg-Based DNA Vaccine for Treating CHB Patients

A preclinical study revealed that DNA constructs that yield HBsAg in vivo effectively induce anti-HBs in HBV TM with very high levels of HBV DNA and HBsAg [59]. Following this, a therapeutic HBV DNA vaccine was employed in CHB patients. Although these patients received antiviral drugs, the outcome was not satisfactory. Fontane et al. concluded that the HBV DNA vaccine neither decreases the risk of relapse in HBV-treated patients nor contains the rate of virological breakthrough [60]. This happened despite viral suppression by NUCs that were used with the DNA vaccine.

### 4.7. Hepatitis B Core Antigen (HBcAg) as an Adjuvant for Treatment of CHB Patients: Preclinical Studies in the Benches and Clinical Trials in Patient’s Bedsides

It became almost clear that the present regimen of immune therapy using polyclonal immune modulators or HBsAg-based vaccine therapy either as monotherapy or as combination therapy with commercially available antiviral drugs would not be able to replace antiviral drugs for treating CHB patients. Accordingly, several approaches were initiated to form a more potent and more efficient regimen of HBV-specific immune therapy. In this line, a vaccine was prepared by the Center for Genetic Engineering and Biotechnology (CIGB), Havana, Cuba. The vaccine is named NASVAC [61]. NASVAC contained 100 micrograms of both HBsAg and HBcAg. First, an animal study was accomplished in Cuba to assess its efficacy. Next, potent immune induction by NASVAC was affirmed by optimizing its usage via the mucosal route [62]. Over time, the installation of cytotoxic T lymphocytes (CTL)-inducing capacity was demonstrated in HBV TM in Japan [63]. After a safety and efficacy study was accomplished in Cuba [64], phase I/II/ and III clinical trials were done with NASVAC in Bangladesh to treat naïve CHB patients with HBV DNA and elevated ALT in the sera [65,66]. The phase III clinical trial compared the safety and efficacy of NASVAC with pegylated IFN. Six weeks after EOT, NASVAC was safer than peg-IFN and endowed with superior efficacy regarding HBV DNA negativity, ALT normalization, and HBeAg seroconversion than peg-IFN. The authors published the follow-up of 2 and 3 years after EOT [67,68]. The data indicate that NASVAC induced sustained HBV negativity and normalized ALT in considerable numbers of patients. NASVAC is now accomplishing a clinical trial in NUC-experienced patients in Japan. The data also unveil the effect of NASVAC on the quantitative HBsAg level [69].

Also, clinical trials have been accomplished by using a therapeutic vaccine containing HBV X and polymerase antigen in addition to HBsAg and HBcAg (GS-4774) with antiviral drugs. In a well-designed phase II study, GS-4774 failed to exhibit the inspiring outcome of this drug in CHB patients [70,71].

### 4.8. Compilation of HBV-Antigen-Specific Immune Therapy for CHB

#### 4.8.1. Limitation of HBsAg-Specific Innovative Immune Therapy for CHB Therapy

HBsAg-based immune therapy (vaccine therapy) was first reported by Pol et al. in 1994 [47]. During the last three decades, several regimens of HBsAg-based immune therapy have been tested in CHB patients. However, none of these therapeutic regimens has been accepted yet for general usage. HBsAg-based immune therapy is safe for CHB patients; however, significant concerns remain about their sustained efficacy. The clinical trials were not designed to respond to this important variable. None of the studies have reported the long-term effects of these therapies. Also, the various trial that used HBsAg-based vaccine with other immune modulators and combination therapy with antiviral drugs have not been designed to provide a new and innovative therapy for clinical usage. Thus, the transformation of research from benches to bedsides was accomplished. But clinical usage for patients with CHB was not planned.

#### 4.8.2. HBsAg Escape Mutant and Innovative Therapy for CHB

Regarding the efficacy and clinical utility of HBsAg-based immune therapy, another important factor should be considered about the emergence of escape mutant of HBsAg due to HBsAg-based vaccination [72]. Vaccine-induced escape mutant of HBV was reported in 1990 by Carman et al. in Italy in the context of prophylactic HB vaccination [73]. Also, the immune-escape mutant has been developed in chronic HBV-infected patients due to the usage of antiviral drugs [74]. It has not been shown if a therapeutic vaccine by HBsAg-based induces such mutant or not. However, this should be checked by clinical trials.

#### 4.8.3. Possible Role of Immune Therapy against Hepatitis Delta Virus

Chronic hepatitis D (CHD) is an outcome of an infection with the HBV and hepatitis D virus (HDV). HBV/HDV coinfection ranges from 1.2% to 25%, however, the prevalence of coinfection varies considerably from place to place [75]. Peg-IFN is mainly used to treat HDVThe role of immune therapeutic agents has not been checked in human HBV/HDV coinfection. However, there remains a possibility of using a therapeutic vaccine with antiviral drugs ac combination therapy.

## 5. Other New and Novel Therapeutic Approaches for Management of CHB

To get proper insights about therapeutic regimens for CHB patients, HBV antigen-specific vaccine therapy included antigens other than HBsAg. A vaccine that contains both HBsAg and HBcAg has inspired considerable optimism about their future usage in CHB patients, not only for its therapeutic efficacy over Peg-IFN but this vaccine was systematically developed after preclinical trial, phase I/II/ and II clinical trials. In addition, the group is publishing data on sustained action of HBsAg/HBcAg-based vaccine [67,68].

Taken together, it appears that HBV antigen-based therapy may be an option of therapeutic choice if multiple HBV-related antigens are used, and a combination of these therapeutic vaccines with other agents may provide a treatment option for CHB.

With the advent of cellular and molecular biology and the realization of the pathogenesis of CHB, it became evident that some measures are needed to accentuate the role of HBV antigen-based therapeutic vaccine in CHB patients. These include variable areas of manipulation in host immunity (Table 4).

### 5.1. Role of Checkpoint Inhibitors for Treating CHB

CHB patients are characterized by chronic exhaustion of CD8-positive T cells, and immune checkpoints play a cardinal role in this phenomenon. Also, studies in animal models indicated that it is possible to revert the efficient function of exhausted T cells if the immune checkpoints can be blocked [76]. This exposed an area of research and clinical application of checkpoint inhibitors as therapeutic options for CHB patients. In the case of chronic HBV infection, PD-1/PDL1 is associated with T cell exhaustion. Also, high levels of PD-1 have been associated with antiviral dysfunction in HBV-specific CD4 cells. In addition to these observations, preclinical studies in woodchuck showed that PD-L1 blocking with NUCs and DNA vaccine treatment boosted virus-specific immunity in chronic HBV infection [77]. PD-1 blockade has also been shown in CHB patients with a therapeutic vaccine [78].

### 5.2. Blocking Negative Impact of Circulating HBsAg on Host Immunity in CHB Patients by HBsAg Inhibitor

Patients with CHB usually harbor large amounts of HBsAg in the sera. Circulating HBsAg is a negative regulator for induction of HBsAg-specific immunity in CHB patients. With this postulation, investigators attempted to reduce HBsAg in the sera in CHB patients. As a result, a therapeutic regimen for reducing HBsAg in the sera was developed. REP 2139, a nucleic acid polymer. REP-2139 is able to block the release of HBsAg from infected hepatocytes. In an open-label, randomized phase II study of HBeAg-negative CHB patients, the safety and efficacy of REP 2139 in combination with tenofovir and Peg-IFN were investigated. REP2139 was found to have some therapeutic efficacy in CHB patients compared to control groups [79,80,81]. However, the safety issue of using REP-2139 must be clarified by conducting viable phase I clinical trials. Also, more studies should be done to optimize its use and explore the mechanism of action.

### 5.3. Engineered T Cells

Functional T cells may be produced by engineering T cells, and these T cells may eliminate HBV-infected hepatocytes. The safety and efficacy of engineered T cells have been assessed in transgenic mice with some safety concerns, but they were capable of controlling HBV replication. The clinical trial with engineered T cells is yet to be accomplished [82,83,84].

### 5.4. Toll-Like Receptor (TLR) Antagonist

Studies have shown that activation of TLR-mediated pathways is capable of suppressing HBV replication and restoration of HBV-specific immunity. Vesatolimod (GS-0620) is an agonist of TLR-7. In a double-blind trial, GS-0620 induced HBV-specific immunity [85]. Selgantolimod is an agonist of TLR-8. It was used as a combination therapy with NUCs. In this trial, HBsAg loss was achieved in 5% of patients with CHB. Thus, there remains considerable optimism about the use of TLR agonists to treat CHB patients [86].

## 6. Summary Based on Past Experiences, Present Realities, and Future Projections about Innovative Therapies for CHB

### 6.1. Past Experiences

Induction of exacerbated immunity was the target of immune therapies in the 1990s for CHB patients. With this concept, polyclonal immune modulators that were available commercially were used in CHB patients. Although these were primarily safe in clinical trials, dose-response studies are lacking. Although these trials were not accomplished to assess the mechanism of action, the agents usually induced increased levels of cytokines. Although following administration of polyclonal immune modulators, reduction of HBV DNA and normalization of ALT were recorded in some trials, and cytokines reduced HBV DNA and normalized ALT in some trials, sustained effects of these agents were neither reported nor study designs planned to provide these. HBsAg-based immune therapy using HBsAg-based prophylactic vaccines was also accomplished from the mid’1990s to the early 2000s. Various protocols were employed to use HBsAg-based therapy, which was also called vaccine therapy. Due to limited therapeutic efficacy, the combination of HBsAg-based vaccine and antiviral drugs was also used in CHB patients to achieve better outcomes for immune therapies in CHB patients. However, convincible studies could not provide substantial optimism about HBsAg-based vaccine therapy or combination therapy of HBsAg-based vaccine plus antiviral agents.

### 6.2. Present Realities

As the usage of polyclonal immune modulators and HBsAg-based vaccines failed to exhibit considerable promise as their therapeutic utilities, immune therapeutic agents containing multiple antigens in addition to HBsAg were produced based on the development of insights about immunopathogenesis of CHB. Among these numerous antigen-based vaccines, HBsAg and HBcAg-based vaccine (NASVAC) and GS-4774, which contain different antigens of HBV, have accomplished clinical trials in CHB patients. The outcome is somehow encouraging, with acceptable safety profiles and moderate efficacy.

### 6.3. Future Projections

In the mean times, some new agents have been documented that are capable of modifying the hepatic mucosal milieu, including different immune dysregulation and dysfunctions. These include check-point inhibitors and blockers of HBsAg production, and clinical trials have affirmed their safety and moderate efficacy. Considering these facts, it seems that a proper regimen of immune therapy may be developed after carefully considering the (1) past experiences, (2) present realities, and (3) future projections.

## Figures and Tables

**Table 1 vaccines-10-00746-t001:** Comprehensive stages of HBV replication.

Variables	Areas of Interference
1. Entry of HBV into hepatocytes	A. Entry inhibitor
2. cccDNA formation in the nucleus of hepatocytes	B. Inhibitor of cccDNA
3. Formation of proteins of the virus	C. RNA interference
4. Role of HBV polymerase regarding HBV replication	D. Nucleoside/Nucleotide analogs
5. Nucleocapsid assembling and pgRNA packaging	E. Core protein allosteric modulators

**Table 2 vaccines-10-00746-t002:** Innovative immune therapy using polyclonal immune modulators.

Agent	Reference
Interleukin-2	[30,31]
Interleukin-12	[32]
Granulocyte-macrophage colony-stimulating factor	[33]
Levamisole	[34]
Thymus humoral factor gamma-2	[35]
Alpha-galactosylceramide	[36]
Thymosine-alpha-1	[37]
Levamisole plus Interferon alpha	[38]

**Table 3 vaccines-10-00746-t003:** HBV antigen-based immune therapy.

Agent	References
HBsAg-antigen-based vaccine therapy	[47,48,49,50,51]
HBsAg antigen plus anti-HBs	[52,53,54]
HBV DNA-based vaccine	[59,60]

**Table 4 vaccines-10-00746-t004:** Immune therapy targets the revival of host immunity.

Agents	Nature of Action	References
1. Checkpoints inhibitor	Blocks PD1-PDL1 to overcome chronic exhaustion of CD8-positive T cells	[76,77,78]
2. Regulator of HBsAg production	Regulating the production of HBsAg with the assumption of overcoming the negative impact of excess HBsAg	[79,80,81]
3. Engineered T cells	Elimination of infected hepatocytes by engineered T cells	[82,83,84]
4. Toll-like receptor agonists	Suppression of HBV replication and restoration of HBV-specific immunity	[85,86]

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
