# Peer review of "Innovative Therapies Targeting the Virus and the Host for Treating Chronic Hepatitis B Virus Infection: From Bench to Bedside"

_vaccines, 2022, doi:10.3390/vaccines10050746_

Round 1
Reviewer 1 Report
It is an interesting review discussing innovative therapies targeting the virus and the host for treating chronic HBV infection. The review includes the following points: a) Therapy targeting HBV replication b) Immune therapy for treating HBV infections c) HBcAg as an adjuvant for treatment of CHB patients. d) Compilation of HBV-antigen-specific immune therapy. e) New approaches for HBV. f) Summary including past, present and future point of view
The review has a good flow and comprehensive. Some suggestions to improve the quality of this manuscript
a) English editing is required for manuscript. There is some misspelling , for example page 3 line 110 in the title: targeting instead of trageting.
b) The authors missed HBS Ag escape mutants, origin, therapy and prevention.
c) The authors need to highlight the impact of the mentioned therapies in case of HBV/HDV coinfections.
Author Response
Response to Reviewer 1
Thank you very much for your encouraging comments and pointing our mistakes about the manuscript. All concerns of the Reviewer have been addressed and the alterations have been shown by YELLOW shading in the revised manuscript.
Responses:
Query of the REVIEWER
- a) English editing is required for manuscript. There is some misspelling, for example page 3 line 110 in the title: targeting instead of trageting.
Response
The manuscript has been checked again. We are sorry for some mistakes. These have been corrected.
The misspellings have been corrected.
Query of the REVIEWER
- b) The authors missed HBS Ag escape mutants, origin, therapy and prevention.
Response:
A subchapter 4.8.2 has been added to address this issue (Line 342-350)
Query
- c) The authors need to highlight the impact of the mentioned therapies in case of HBV/HDV coinfections.
Response
A subchapter 4.8.2 has been added to address this issue (Line 351-359).

Reviewer 2 Report
In this review paper, Akbar et al. summarized recent findings in novel therapeutic approaches to HBV infection, especially focusing on immunotherapy including vaccines. The manuscript is written almost well, but there are several concerns to be addressed as below.
- Line 48, “a double-edged sword” – the meaning of this phrase seems not to fit in this sentence.
- Line 51, “chronic hepatitis CHB (CHB)” – please correct.
- Lines 63-64, “NUCs have been reported to contain liver damage to certain levels in some CHB patients” – clinically, in general, hepatotoxicity of NUCs is negligible. Please reconsider the description.
- Lines 64-66, “However, final analyses indicate that NUCs could not stand the test of time as therapeutic choices for millions of CHB patients” – it is unclear what this sentence means.
- Line 73, please remove “(HBV)”.
- Line 83, “the virus is not directly oncogenic” – I disagree with this description. HBV proteins, such as HBx, have been reported to be oncogenic.
- Line 116, “blocking RNA polymerase activity” should be modified to “blocking DNA polymerase activity”.
- Table 1 – “D. Nucleoside/Nucleotise analogs” should be corrected to “D. Nucleoside/Nucleotide analogs”.
- Table 1 – “5. Nucleoside assembling and pgRNA packaging” should be corrected to “5. Nucleocapsid assembling and pgRNA packaging”.
- Section 3.1. Entry inhibitor – myrcludex-B has been approved with a name of Bulevertide for medical use in the European Union in July 2020 for patients with chronic hepatitis B/D coinfection. Please mention this point.
- Table 2 – “Interleukic-2” should be corrected to “Interleukin-2”.
- Table 2 – please remove a comma after “Levamisole”.
- Section 4. Innovative immune therapy for treating CHB – please mention TLR agonist, which are being studied on some clinical trials.
- Table 3 – please correct “HBsAg-antigen-base vaccine therapy” to “HBsAg-antigen-based vaccine therapy”. Also, please show appropriate references for each agent.
- Sections 5 and 6 – these are a part of immune therapy and should be included in the section 4 “Innovative immune therapy for treating CHB”.
- Table 4 – “Immun therapy” should be corrected to “Immune therapy”.
- Table 4 – the information of this table is a little. Please show the further detail of each agent with appropriate references.
- Lines 328-329 – “HBsAg-antigen immune therapy” should be corrected to “HBsAg immune therapy”. Also, please add a reference on this sentence.
- Lines 343-344, “In addition, the group is publishing data on sustained action of HBsAg/HBcAg-based vaccine” – please add an appropriate reference.
- Throughout the manuscript, many of the reference numbers seem to be incorrect. Please fix them.
Author Response
Response to Reviewer 2
Thank you very much for your encouraging comments and pointing our mistakes about the manuscript. All concerns of the Reviewer have been addressed and the alterations have been shown by YELLOW shading in the revised manuscript.
- Line 48, “a double-edged sword” – the meaning of this phrase seems not to fit in this sentence.
Response: This part has been omitted.
- Line 51, “chronic hepatitis CHB (CHB)” – please correct.
Response: This has been corrected (Line 50).
- Lines 63-64, “NUCs have been reported to contain liver damage to certain levels in some CHB patients” – clinically, in general, hepatotoxicity of NUCs is negligible. Please reconsider the description.
Response: This sentence has been changed, as advised (Line 63-68).
- Lines 64-66, “However, final analyses indicate that NUCs could not stand the test of time as therapeutic choices for millions of CHB patients” – it is unclear what this sentence means.
Response: This sentence has been omitted (Line 63-68).
- Line 73, please remove “(HBV)”.
Response: This has been removed (Line 72)
- Line 83, “the virus is not directly oncogenic” – I disagree with this description. HBV proteins, such as HBx, have been reported to be oncogenic.
Response: Considering the oncogenecity of HBx protein, the sentence has been omitted.
- Line 116, “blocking RNA polymerase activity” should be modified to “blocking DNA polymerase activity”.
Response: This has been done (Line 116)
- Table 1 – “D. Nucleoside/Nucleotise analogs” should be corrected to “D. Nucleoside/Nucleotide analogs”.
Response: This has been corrected (Table 1)
- Table 1 – “5. Nucleoside assembling and pgRNA packaging” should be corrected to “5. Nucleocapsid assembling and pgRNA packaging”.
Response: This has been corrected (Table 1).
- Section 3.1. Entry inhibitor – myrcludex-B has been approved with a name of Bulevertide for medical use in the European Union in July 2020 for patients with chronic hepatitis B/D coinfection. Please mention this point.
Response: This has been modified. (Line 129-132)
- Table 2 – “Interleukic-2” should be corrected to “Interleukin-2”.
Response: This has been corrected (Table 2)
- Table 2 – please remove a comma after “Levamisole”.
Response: This has been removed.
- Section 4. Innovative immune therapy for treating CHB –please mention TLR agonist, which are being studied on some clinical trials.
Response: This has been shown in a new subchapter 5.4 (Line
- Table 3 – please correct “HBsAg-antigen-base vaccine therapy” to “HBsAg-antigen-based vaccine therapy”. Also, please show appropriate references for each agent.
Response; The mistake has been corrected. References have been added.
- Sections 5 and 6 – these are a part of immune therapy and should be included in the section 4 “Innovative immune therapy for treating CHB”.
Response: This has been done in a section 5.4. (Line 408-414)
- Table 4 – “Immun therapy” should be corrected to “Immune therapy”.
Response: This has been corrected.
- Table 4 – the information of this table is a little. Please show the further detail of each agent with appropriate references.
Response: This has been done
- Lines 328-329 – “HBsAg-antigen immune therapy” should be corrected to “HBsAg immune therapy”. Also, please add a reference on this sentence.
Response: This has been corrected. Reference has also been added. (Line 329)
- Lines 343-344, “In addition, the group is publishing data on sustained action of HBsAg/HBcAg-based vaccine” – please add an appropriate reference.
Response: References have been added (Line 367).
- Throughout the manuscript, many of the reference numbers seem to be incorrect. Please fix them.
Response: References have been checked and all these have been marked by yellow shading)

Round 2
Reviewer 1 Report
The authors addressed my suggestions
Author Response
Response to Reviewer 1
Query:
The authors addressed my suggestions
Response
Thanks for consideration

Reviewer 2 Report
The authors modified the manuscript substantially but there are still some issues to be fixed as below.
- References has not been added yet in Table 3. Please add them in a same manner as Table 2.
- The new Table 4 seems to be out of a general style of table. Please reconsider the style of Table 4 for easy understanding.
- A reference has not been added yet in a sentence (lines 331-332) “HBsAg-based immune therapy (vaccine therapy) was first reported by Pol et al. in 1994.”
- Lines 349-350 – “At has not been shown if therapeutic vaccine by HBsAg-based induce such mutant or not” should be corrected to “It has not been…”
- Lines 357-360 – myrcludex-B is not an immune therapy and the description of this drug should not be included in this section. Are there any reports of HBsAg-based immune therapy in patients with HBV/HDV coinfection? Please include this point in this section.
Author Response
The authors modified the manuscript substantially but there are still some issues to be fixed as below.
- Query:
References have not been added yet in Table 3. Please add them in the same manner as Table 2.
Response:
References have been added in Table 3, as shown in Table 2. This has been done according to the recommendation of the Reviewer.
- Query:
The new Table 4 seems to be out of a general style of table. Please reconsider the style of Table 4 for easy understanding.
Response:
The Table has been modified according to the recommendation of the Reviewer (Table 4).
- Query:
A reference has not been added yet in a sentence (lines 331-332) “HBsAg-based immune therapy (vaccine therapy) was first reported by Pol et al. in 1994.”
Response: A reference, reference 47 has been added
- Query:
Lines 349-350 – “At has not been shown if therapeutic vaccine by HBsAg-based induces such mutant or not” should be corrected to “It has not been…”
Response
I am sorry for the mistake. This has been corrected.
- Query
Lines 357-360 – myrcludex-B is not an immune therapy and the description of this drug should not be included in this section. Are there any reports of HBsAg-based immune therapy in patients with HBV/HDV coinfection? Please include this point in this section.
Response
Thanks for your critical observation about the entry inhibitor. Unfortunately, I could not find any study on HBV/HDV coinfection with HBsAg-based immune therapy. The sentence has been optimized.
